# The Control of the Crossover Localization in Allium

**DOI:** 10.3390/ijms24087066

**Published:** 2023-04-11

**Authors:** Natalia Kudryavtseva, Aleksey Ermolaev, Anton Pivovarov, Sergey Simanovsky, Sergey Odintsov, Ludmila Khrustaleva

**Affiliations:** 1All-Russian Research Institute of Agricultural Biotechnology, 42 Timiryazevskaya Str., Moscow 127550, Russia; 2Center of Molecular Biotechnology, Russian State Agrarian University—Moscow Timiryazev Agricultural Academy, 49 Timiryazevskaya Str., Moscow 127550, Russia; 3Severtsov Institute of Ecology and Evolution, Russian Academy of Sciences, 33 Leninsky Prosp., Moscow 119071, Russia

**Keywords:** chiasmata localization, MUS81, MLH1, ASY1, ZYP1, Allium, GISH

## Abstract

Meiotic crossovers/chiasmata are not randomly distributed and strictly controlled. The mechanisms behind crossover (CO) patterning remain largely unknown. In *Allium cepa*, as in the vast majority of plants and animals, COs predominantly occur in the distal 2/3 of the chromosome arm, while in *Allium fistulosum* they are strictly localized in the proximal region. We investigated the factors that may contribute to the pattern of COs in *A. cepa*, *A. fistulosum* and their F${}_{1}$ diploid (2*n* = 2*x* = 8C + 8F) and F${}_{1}$ triploid (2*n* = 3*x* = 16F + 8C) hybrids. The genome structure of F${}_{1}$ hybrids was confirmed using genomic *in situ* hybridization (GISH). The analysis of bivalents in the pollen mother cells (PMCs) of the F${}_{1}$ triploid hybrid showed a significant shift in the localization of COs to the distal and interstitial regions. In F${}_{1}$ diploid hybrid, the COs localization was predominantly the same as that of the *A. cepa* parent. We found no differences in the assembly and disassembly of ASY1 and ZYP1 in PMCs between *A. cepa* and *A. fistulosum*, while F${}_{1}$ diploid hybrid showed a delay in chromosome pairing and a partial absence of synapsis in paired chromosomes. Immunolabeling of MLH1 (class I COs) and MUS81 (class II COs) proteins showed a significant difference in the class I/II CO ratio between *A. fistulosum* (50%:50%) and *A. cepa* (73%:27%). The MLH1:MUS81 ratio at the homeologous synapsis of F${}_{1}$ diploid hybrid (70%:30%) was the most similar to that of the *A. cepa* parent. F${}_{1}$ triploid hybrid at the *A. fistulosum* homologous synapsis showed a significant increase in MLH1:MUS81 ratio (60%:40%) compared to the *A. fistulosum* parent. The results suggest possible genetic control of CO localization. Other factors affecting the distribution of COs are discussed.

## 1. Introduction

Crossover (recombination) is the exchange of DNA between maternal and paternal chromosomes during meiosis with the formation of new combinations of genetic material inherited in the next generation. Meiotic crossovers (COs) in plants are not randomly distributed along the length of individual chromosomes, and are tightly regulated [1,2,3,4,5]. COs often occur in the distal 2/3 of the chromosome arm, while suppression of recombination is observed in the proximal centromere regions in most eukaryotes [6,7,8]. The exceptions are several species with COs strictly localized in the proximal region adjacent to the centromere, and among them are *Allium fistulosum* [9,10,11]. The cytological manifestation of COs is chiasmata, the position of which determines the configuration of bivalents at metaphase I of meiosis. Previously, it was thought that distal chiasmata may have originated in more central regions of chromosomes and migrated by a process called terminalization to distal regions, but now it is known that chiasmata do not terminate, and therefore their location reflects their real place of origin [2,12,13]. COs promote accurate segregation of homologous chromosomes, and this is a key moment in the formation of the chromosome set of gametes. To prevent the appearance of non-exchange chromosomes, COs are strictly controlled. Obligating one CO per bivalent is a necessary condition for the correct segregation of chromosomes in meiosis [14,15,16]. COs are initiated by DNA double-strand breaks (DSBs) formed by the topoisomerase-like protein SPO11, along with a cohort of partner subunits [3]. The number of DSBs occurs in excess compared to crossovers, likely to ensure that at least one crossover will be formed per homolog pair [17,18,19,20]. DSBs that do not become crossovers are repaired to give non-crossovers. The phenomenon of interference discovered at the beginning of the last century [21,22] is one of the pieces of evidence indicating the existence of control over the formation of COs. The term "interference" refers to the observation that a CO in one region of a chromosome interferes with another CO being formed nearby [17].

The meiotic recombination pathway begins with each chromosome binding to a protein axis that includes ASY1, the lateral element of the synaptonemal complex (SC), at leptotene [23]. In *asy* mutants of *Arabidopsis thaliana* in absence of ASY1, telomere-led recombination becomes dominant, limiting spaced crossover formation along the chromosomes [24]. The ZYP1 protein, the component of a central element of SC, plays the role of a molecular zipper to bring homologous chromosomes in close apposition and synapsis formation [25,26,27]. Recent studies in plants have shown that ZYP1 may be a key player in controlling the number and location of meiotic crossovers [25,28].

There are two major pathways for meiotic crossing-over. One pathway requires many proteins, including the ZMM group (ZYP1-4, MSH4-5, MER3) and the MutL homolog 1 (MLH1)/MLH3 complex [29,30]. COs produced by this pathway, named class I COs, are subject to interference. The other pathway does not require ZMM proteins, but depends on the MUS81/Mms4 endonuclease complex in budding yeast [31,32] and the MUS81/EME1 complex in plants [33,34,35]. MUS81 indicates COs that may be formed by the processing of non-Holliday junction (HJ) intermediates and involved in a secondary subset of meiotic crossovers that are interference insensitive [33,36,37]. COs produced by this pathway are named class II COs. It has recently been found in *A. thaliana* that class II COs can be generated by at least two parallel pathways which depend on either the structure-specific endonuclease AtMUS81, or a homolog of Fanconi Anemia Complementation Group D2 (AtFANCD2) that promotes noninterfering COs [38].

For cytological study, immunolabeling of MLH1 and MUS81 is usually used to mark designated sites of class I and II COs in plants [39,40,41,42,43,44]. The frequency ratio of MLH1 and MUS81 ranged between 75–85% (MLH1) and 15–25% (MUS81), observed on *A. thaliana* [35], *Solanum lycopersicum* [41,42] and allotetraploid (AABB) durum wheat (*Triticum turgidum* subsp. durum) [43]. Class II COs predominantly occur in pericentromeric heterochromatin, as shown on the tomato [42]. There is a complex interaction between class I and II pathways to generate COs. A study of meiosis in mice has shown that loss of MUS81 results in a compensatory increase in MLH1-tagged nascent COs as a result of increased class I COs [45]. For the first time in plant cytogenetics, Anderson and coauthors [42] reported that classes I and II of COs are not independent. Using a combination of immunofluorescent localization of meiotic proteins and distribution of late recombination nodules (LNs) on the tomato SCs, the authors showed that classes I and II have different recombination profiles along the chromosome, and the two crossing-over pathways interact with each other.

All cytogenetic studies using immunodetection of proteins involved in the recombination and assembly of the synaptonemal complex were carried out on plants with distal chiasmata [35,41,42,43]. To our knowledge, plants with strictly localized chiasmata in the pericentromeric region, such as *A. fistulosum*, have never been studied in this aspect. However, this phenomenon has been exciting geneticists and cytologists since 1935. Emsweller and Jones [10] were intrigued by the different localization of chiasmata in two closely related Allium species. In *A. cepa*, the chiasmata are localized in the distal and interstitial regions and appear as ring bivalents, while in *A. fistulosum*, on the contrary, the chiasmata are strictly located in the proximal region and appear as cross-bivalents. Analyzing the meiosis of the F${}_{1}$ hybrid between *A. cepa* and *A. fistulosum*, the authors found that all the bivalents were rings, as in *A. cepa*. It has been suggested that *A. cepa* possesses a dominant gene, and *A. fistulosum* a recessive gene, and possible genetic control of the distribution of chiasmata has been hypothesized. Later analysis of the synaptonemal complex in F${}_{1}$ hybrid between *A. cepa* and *A. fistulosum* revealed a synapsis disorder in the centromere region, which could be the cause of a significant decrease in COs in the proximal region [46]. *A. fistulosum* is a rich reservoir of desirable traits for improving the bulb onion (*A. cepa*) gene pool [47,48,49]. Knowledge of CO location control can help to successfully introduce desirable traits of *A. fistulosum* into onions.

In this study, we employed immunodetection of proteins, involved in CO and SC formation (MLH1, MUS81, ASY1 and ZYP1) and analysis of chiasmata localization in two closely related Allium species, showing highly contrasting patterns of chiasmata distribution, and F${}_{1}$ hybrids between them. Using genomic *in situ* hybridization (GISH), F${}_{1}$ (*A. cepa*× *A. fistulosum*) hybrids with diploid (2*n* = 2*x* = 8F + 8C) and triploid (2*n* = 3*x* = 16F + 8C) sets of chromosomes were revealed. The availability of such a unique plant material allowed us to study the localization of chiasmata and the ratio of MLH1 and MUS81 on homeologous pairs *A. cepa*/*A. fistulosum* and on homologous pairs of *A. fistulosum* in the presence of a haploid set of *A. cepa* at pachytene in pollen mother cells (PMCs). The results suggest possible genetic control of CO localization.

## 2. Results

### 2.1. GISH Analysis of Genomic Structure of F${}_{1}$ Hybrids (A. cepa × A. fistulosum)

Scoring of the chromosome number in F${}_{1}$ hybrids on the mitotic metaphase spreads revealed individual plants with 16 and 24 chromosomes. GISH analysis showed that the F${}_{1}$ hybrid with 16 chromosomes displayed 8 chromosomes of *A. cepa* (8C) and 8 chromosomes of *A. fistulosum* (8F) (Figure 1a). Karyotype analysis showed the presence of complete haploid sets of both parental species: 2*n* = 2*x* = 8F + 8C (Figure 1a${}^{\prime }$). GISH analysis of the F${}_{1}$ triploid with 24 chromosomes displayed 8 chromosomes of *A. cepa* (8C) and 16 chromosomes of *A. fistulosum* (16F) (Figure 1b). Karyotype analysis showed the presence of a complete haploid set of *A. cepa* and complete diploid sets of *A. fistulosum* chromosomes: 2*n* = 3*x* = 16F + 8C (Figure 1b${}^{\prime }$). The GISH result indicates the formation of unreduced 2n-gametes during microspogenesis in *A. fistulosum*. The F${}_{1}$ triploid hybrid is a unique plant material for studying the effect of the *A. cepa* haploid genome on the configuration of the *A. fistulosum* homologous bivalents, assembly and disassembly of the synaptonemal complex, and MLH1 and MUS81 protein immunodetection.

### 2.2. The Chiasmata Distribution in PMCs at Metaphase I

The acetocarmine stained squash preparations of pollen mother cells (PMCs) at metaphase I of *A. cepa* and *A. fistulosum*, and their F${}_{1}$ diploid and triploid hybrids were analyzed.

In *A. cepa*, chiasmata occurred mainly in distal and interstitial bivalent regions and are very rare in proximal regions (Table 1). Chiasmata localized in the distal 2/3 of the chromosome arm accounted for 98.2%, and in the proximal 1/3 there were only 1.8%. The majority of metaphases contained eight bivalents with two distal chiasmata. Bivalents with three chiasmata were also common, especially in long chromosomes, in which two distal and one interstitial chiasmata often occurred (Figure 2a). The frequency of chiasmata per cell was 19.1.

The extreme localization of chiasmata to the proximal regions adjacently to the centromere of the *A. fistulosum* bivalents was evident, although occasionally, distal and interstitial chiasmata and univalents were found (Figure 2b). Chiasmata localized in the proximal 1/3 of the chromosome arm accounted for 97.1%, and in the distal 2/3 there were only 2.9% (Table 1). Open bivalents and univalents have been observed, which was reflected in the frequency COs per cell—15.4.

In diploid F${}_{1}$ hybrid (*A. cepa*×*A. fistulosum*), carrying eight chromosomes of the *A. cepa* female parental species and eight chromosomes of the *A. fistulosum* male parental species, the chiasmata distribution was most similar to that of the *A. cepa* parent. No bivalents were observed with only proximally localized chiasmata as in *A. fistulosum*. The bivalents possessed chiasmata mostly in distal and interstitial regions. Homeologous chromosomes conjugated during prophase I of meiosis and formed chiasmata, indicating sufficient homology despite the difference in genome size between *A. fistulosum* and *A. cepa*. The genome size of *A. cepa* (1C = 16.4 Gb, [50]) has 4.8 Gb more DNA and correspondingly larger chromosomes than *A. fistulosum* (1C = 11.6 Gb, [51]). We have observed heteromorphic bivalents, open bivalents with one chiasma and univalent pair of unequal size (Figure 2c). In the F${}_{1}$ hybrid, 47.7% of the chiasmata were located in distal regions of bivalents and 41.3% in interstitial ones, which is considerably higher than that of *A. fistulosum* (1.7% and 1.2%, correspondingly).

In the F${}_{1}$ triploid hybrid, bivalents are mainly formed due to the pairing of homologous chromosomes of *A. fistulosum* in the presence of a complete haploid set of *A. cepa* chromosomes. Surprisingly, homomorphic bivalents of *A. fistulosum* showed pronounced chiasmata localization in the distal 2/3 of the chromosome arm (in distal region—34.5% and interstitial region—26.0%), and only 39.5% in the proximal region compared to *A. fistulosum*—97.1% (Table 1). Trivalents formed by two homologous chromosomes of *A. fistulosum* and one homeologous chromosome of *A. cepa* were observed very rarely (Appendix A).

The data of chiasmata distribution in *A. cepa*, *A. fistulosum* and their F${}_{1}$ diploid and triploid hybrids presented in Table 1 were analyzed using a $\chi^{2}$ statistic. Analysis of the contingency table containing the number of chiasmata on different locations in analyzed plant samples (Table 1) showed the presence of statistically significant differences in chiasmata locations among analyzed samples ($\chi^{2}$ = 1966.4, df = 6, *p* value < 0.001, Cramer’s V = 0.62). Pairwise comparisons showed the presence of statistically significant differences in chiasmata types between analyzed samples in all pairs (Table 2). Pairwise Cramer’s V values indicate that *A. cepa* and *A. fistulosum* have the most different patterns of chiasmata location (Cramer’s V = 0.95), while F${}_{1}$ diploid and triploid hybrids’ chiasmata location patterns are the least different (Cramer’s V = 0.24). Both F${}_{1}$ diploid and triploid hybrids’ chiasmata location patterns are more similar to *A. cepa* (Cramer’s V = 0.34 for F${}_{1}$ diploid hybrid and Cramer’s V = 0.49 for triploid F${}_{1}$ hybrid) rather than *A. fistulosum* (Cramer’s V = 0.87 for F${}_{1}$ diploid hybrid and Cramer’s V = 0.72 for triploid F${}_{1}$ hybrid), which is consistent with the hypothesis about dominant gene-controlling chiasmata distributions in *A. cepa* suggested by Emsweller and Jones [10].

### 2.3. Behavior of ASY1 and ZYP1 during SC Assembly and Disassembly in A. cepa, A. fistulosum, and their F${}_{1}$ Diploid and Triploid Hybrids

Meiotic COs occur in association with synaptonemal complexes (SCs) that connect paired homologous chromosomes during pachytene. In *A. thaliana*, it was shown that SC have a key role in regulating CO numbers and distributions [24,25].

The dynamics of the SC during prophase I were monitored by immunolocalization of ASY1 (lateral elements of SC) and ZYP1 (transverse filaments of SC) in *A. cepa*, *A. fistulosum* and their F${}_{1}$ diploid and triploid hybrids. A comparative analysis of ASY1 and ZYP1 behavior in the hybrids and their parental species was performed.

In *A. cepa* at leptotene, ASY1 loaded on the meiotic chromosome axes, while ZYP1 began to be loaded along the synapsis sites. At zygotene, ASY1 was detected as single thin tracks that corresponded to unpaired chromosome axes, while ZYP1 tracks became longer along synapsed chromosome axes. At pachytene, ZYP1 tracks have fully formed along the synapsed homologous chromosomes, while only a few unpaired regions can be identified by some weak ASY1 signals or as a remaining diffuse signal. During diplotene, ZYP1 undergoes degradation, indicating the beginning of disassembly of synaptonemal complexes. At diakinesis, ZYP1 signals appear as ball-like structures (Figure 3 and Appendix A).

In *A. fistulosum*, the behavior of ASY1 and ZYP1 was similar to that of *A. cepa*. We found no differences in the assembly and disassembly of these proteins between *A. cepa* and *A. fistulosum* (Figure 3 and Appendix A). However, ASY1 and ZYP1 of both *A. cepa* and *A. fistulosum* showed the peculiarities of loading and unloading during prophase I of meiosis. ZYP1 loaded only at the pairing regions of chromosomes, while ASY1 signals disappeared from them. Thus, the tripartite SC structure, combining two ASY1 linear structures and one ZYP1 core, has never been observed in *A. cepa*, *A. fistulosum*, and their F${}_{1}$ hybrids (Figure 3a–d). The same ZYP1 loading pattern was shown in maize, wheat and rice, characterized by the absence of the tripartite SC structure [52,53,54,55]. In contrast, the tripartite SC structure is formed at pachytene in Arabidopsis, barley and rye [56,57,58,59].

In the F${}_{1}$ diploid hybrid, synapsis was formed between homeologous chromosomes. At leptotene, comparative analysis of the F${}_{1}$ diploid hybrid and parental species revealed no difference in ASY1 loading (Figure 3 and Appendix A). ASY1 was detected as single thin tracks forming along unpaired chromosome axes, while ZYP1 assembly delay was found, as evidenced by the lower number of ZYP1 tracks compared to parental species. At zygotene, a number of unpaired chromosomes with loaded ASY1 remained due to the above-mentioned delay in homeologous synapsis, while ZYP1 continued to load along the synapsed chromosome axes. In the middle of the pachytene, long, thick ASY1 tracks were observed, indicating that long regions of homeologous chromosomes had not yet synapsed. The number of ZYP1 tracks was lower compared to the parent species. At late pachytene, synapsis formation was accomplished, and long ZYP1 tracks were observed. Furthermore, the superimposition of DAPI and ZYP1 images revealed a partial absence of synapsis in paired chromosomes (Appendix A). These observations coincided with our data of chiasma analysis at metaphase I. As a result of the lack of synapsis in several sites between homeologous chromosomes, univalents were observed (Figure 2c). At diplotene and diakinesis, the disassembly of the synaptonemal complex did not differ from the parental species.

Comparison between the F${}_{1}$ triploid hybrid and parental species revealed no difference in ASY1 and ZYP1 loading at leptotene (Figure 3 and Appendix A). This observation can be explained by the presence of a diploid set of *A. fistulosum* chromosomes that readily form homologous synapsis. At zygotene, long tracks of ASY1 were observed, which is probably associated with the presence of an unpaired set of *A. cepa* chromosomes (Appendix A), and ZYP1 long tracks were more likely formed between homologous chromosomes of *A. fistulosum*. At pachytene, ASY1 long tracks appeared along with ASY1 short tracks, which probably derived from abnormal synapsis of a single *A. cepa* chromosome with two homologous chromosomes of *A. fistulosum* (Appendix A). The ZYP1 long tracks coincided with paired DAPI stained chromosomes that indicates complete synapsis of homologous chromosomes of *A. fistulosum* (Figure 3 and Appendix A). At diplotene and diakinesis, the disassembly of the synaptonemal complex did not differ from the parental species.

### 2.4. Immunocytological Detection of MLH1 and MUS81 on Pachytene Chromosomes of A. cepa and A. fistulosum and their F${}_{1}$ Diploid and Triploid Hybrids

In order to lift the veil over the phenomenon of proximal chiasmata location, we (1) studied the features of ASY1 and ZYP1 during SC assembly and disassembly, and (2) identified the pathways of chiasmata (COs) origin via immunolocalization of MLH1 and MUS81 in *A. cepa*, *A. fistulosum*, and their F${}_{1}$ diploid and triploid hybrids.

Scoring of the chiasmata distribution showed significant differences in *A. cepa* and *A. fistulosum*. In the F${}_{1}$ diploid hybrid, chiasmata were predominantly located in distal and interstitial regions of homeologous bivalents, and in F${}_{1}$ triploid hybrid a significant distal shift in homologous bivalents was observed. Chiasmata only indicates that a CO has occurred, not its class type. The MLH1 protein involved in meiotic recombination is commonly used as a marker for class I COs that are susceptible to interference, and another protein, MUS81, for class II COs that are indifferent to interference [30,32,42,43,60]. In order to find out the pathway of origin of COs, we performed immunolabelling of MLH1 and MUS81. For this purpose, polyclonal antibodies against the *A. cepa* protein sequences were prepared. We have counted the number of MLH1 and MUS81 foci per cell on ZYP1 tracks in prophase I of PMCs. Recently, it was shown that ZYP1 is required for class I CO formation in wild types of *A. thaliana* [26] and *Brassica rapa* [61]. However, in *zyp* mutants of *A. thaliana*, ZYP1 is dispensable for class I CO formation [26,35]. Immunolocalization of MUS81 in wild types of *A. thaliana* and wheat showed its association with ZYP1 [35,43,62]. Such studies have never been performed on onions.

In order to clearly distinguish the MLH1 and MUS81 signals from the background, only the fluorescent signal located on the ZYP1 track was considered as a true signal. Dual-color immunodetection of MLH1/ZYP1 and MUS81/ZYP1 was performed on different preparations of pachytene chromosomes. We could not carry out simultaneous detection of MLH1, MUS81 and ZYP1 on one preparation due to the unavailability of the three-color antibody system. Preparation of pachytene chromosomes was carried out on a population of PMCs from a single plant in each variant at the same time for MLH1 and MUS81. The MLH1 and MUS81 foci data per cell, obtained by counting on different slides, did not reveal significant differences between an artificial population (see Section 4) in which both MLH1 and MUS81 were expected to be counted in the same cell (Appendix A).

In *A. cepa*, dual immunodetection of MLH1 in conjunction with ZYP1 revealed that MLH1 formed an average of 11.1 foci per cell. The average number of MUS81 foci associated with ZYP1 was 4.1 (Table 3, Figure 4). The total number of MLH1 and MUS81 foci per cell was 15.2, which is less than the frequency of COs (19.1) established at metaphase I (Table 1). This minor discrepancy may be due to technical limitations related to the accessibility of antibodies to target proteins on the onion pachytene chromosomes. Although it is possible that there is a third pathway of CO formation, the recombinant proteins were not used as a marker for COs in this study. It has recently been shown in *A. thaliana* that AtFANCD2 is required for meiotic homologous recombination, promoting noninterfering COs in a MUS81-independent manner [62]. The FANCD2 protein is monoubiquinated in response to DNA damage, resulting in its localization to nuclear foci with other proteins (BRCA1 and BRCA2) involved in homology-directed DNA repair [63].

We have demonstrated via immunodetection of meiotic proteins during prophase I of *A. cepa* that MLH1 was essential for the majority of foci per cell (73%) compared to MUS81 (27%). This result is similar to the frequency ratio of MLH1 and MUS81 observed on *A. thaliana* [35], *Solanum lycopersicum* [41,42] and allotetraploid (AABB) durum wheat (*Triticum turgidum* subsp. durum) [43].

Dual immunodetection of MLH1 and MUS81 in *A. fistulosum* revealed an approximately equal mean number of foci per cell—6.5 both for MLH1 and MUS81 (Table 3, Figure 4). The number of MLH1 foci per cell was 4.6 less in *A. fistulosum* compared to *A. cepa* (*p* value <0.001, Appendix A) and 2.4 more for MUS81 (*p* value <0.001, Appendix A). The ratio of MLH1 and MUS81 foci per cell was 50%:50%, which is significantly different from *A. cepa* (73%:27%). The observed difference in ratio of MLH1 and MUS81 foci is in agreement with the distribution of chiasmata in these species. It is possible that MLH1, which is involved in the interference-dependent COs formation, has a limited ability to operate on the specifically organized *A. fistulosum* chromatin at the genomic and epigenomic levels. Under these circumstances, MUS81 should predominantly recruit for the interference-free CO formation to guarantee obligate recombination. This observation points to interference between class I and II COs that coincides with previously reported data on tomato [42].

In the F${}_{1}$ diploid hybrid, we observed a mean of 8.5 MLH1 and 3.6 MUS81 foci per cell (Table 3, Figure 4). All detected signals of MLH1 and MUS81 were observed on the ZYP1 tracks formed between homeologous chromosomes *A. cepa*/*A. fistulosum*. Immunocytological analysis of the ZYP1 assembly in this study revealed the lack of ZYP1 in the short segments of paired homeologous chromosomes in F${}_{1}$ diploid hybrid. In order to identify MLH1 and MUS81 signals at sites with missed ZYP1 on paired chromosomes, signal scoring was also performed on merged images of MLH1/ZYP1 and MUS81/ZYP1 with DAPI-stained chromatids. We did not detect any signal in ZYP1 missing sites. The total number of MLH1 and MUS81 foci per cell was 12.1, which is less than in parental species due to univalents and open bivalents, that were observed at metaphase I (Figure 2). However, the ratio of MLH1/MUS81 per cell (70%:30%) was almost equal to that of *A. cepa* (73%:27%). The result corresponds to the location of the chiasmata predominantly in the distal and proximal regions (Table 1, Figure 2). The observation of an *A. cepa* type of MLH1/MUS81 coincides with a predominantly *A. cepa* type of chiasmata location in this and previous studies [46].

In the F${}_{1}$ triploid hybrid, we observed a mean of 8.5 MLH1 and 5.7 MUS81 foci per cell (Table 3, Figure 5). The total number of MLH1 and MUS81 foci per cell was 14.2 (Table 1). The frequency ratio of MLH1 and MUS81 is 60%:40%. In this F${}_{1}$ triploid hybrid, CO formation occurs between the homologous chromosomes of *A. fistulosum*, and we may expect the MLH1/MUS81 ratio to be similar to *A. fistulosum* (50%:50%). However, the presence of the haploid set of *A. cepa* chromosomes modified the MLH1/MUS81 ratio. The increased number of MLH1 foci associated with an additional gene dosage in the F${}_{1}$ triploid hybrid (*p* value < 0.001, Appendix A) is probably one of the players responsible for the distal shift of chiasmata (Table 1, Figure 2).

## 3. Discussion

In this study, we set out to test the hypothesis of possible genetic control of COs/chiasmata distribution, and tried to determine the factors underlying the highly contrasting patterns of chiasmata localization in two closely related Allium species. To investigate this phenomenon, we performed immunological and cytogenetic assays of CO formation components that are detectable in the prophase I and metaphase I stages of meiosis. Additionally, the behavior of ASY1 and ZYP1 during the assembly and disassembly of the SCs was studied. We used a unique plant material, which made it possible to study the formation of COs on heteromorphic (nonhomologous) and homomorphic (homologous) meiotic pairs of chromosomes. A distal shift of chiasmata on the *A. fistulosum* homologous chromosome pairs in the presence of the *A. cepa* genome suggests genetic control of CO localization. Furthermore, this study has provided new insights into the interaction of two pathways of CO formation to maintain total CO levels per nucleus in heteromorphic and homomorphic meiotic pairs in interspecific hybrids. New knowledge of SC assembly/disassembly was obtained that has not previously been reported for Alliums.

### 3.1. Chromatin, Genome and the Highly Contrasting Chiasmata Localization in A. cepa and A. fistulosum

Studies of CO patterns in many organisms have shown high levels of COs in distal euchromatin and low levels of COs in pericentric heterochromatin, whether analyzed using chiasmata, genetic maps, genotyped recombinant inbred lines, recombinant nodules, or MLH1 foci [7,24,41,42,64,65,66]. It should be noted that all of these studies were carried out on organisms with a distal distribution of COs. Species with strictly proximally localized COs are rare among plants and animals. To our knowledge, among the plant species studied for recombination distribution, only four species are known with extreme localization of chiasmata, namely, *Allium fistulosum* [10], *Allium porrum* [67,68], and *Allium altaicum* [11,69], which are very close phylogenetically, and *Fritillaria meleagris* [70], belonging to the Liliaceae family. All four species belong to clade Liliopsida, subclass Petrosaviidae. The proximal localization of the chiasmata in *Fritillaria meleagris* is explained by an association between chiasmata localization and restricted synapsis [2], which is not typical for the Allium species [46]. What do we know about chromatin organization in *A. fistulosum* and *A. cepa*? The localization of constitutive heterochromatin along the entire chromosome is usually observed using differential C-band staining on mitotic metaphase chromosomes. Comparative analysis of the C-band pattern in *A. fistulosum* and *A. cepa* did not reveal significant differences [11]. In both species, all chromosomes possess terminal C-bands which are larger in *A. fistulosum* than those found in *A. cepa*, and centromeric heterochromatin is present in all the chromosomes in the form of very thin bands sometimes seen as paired dots. Comparing the C-band pattern with the distribution of chiasmata in species of the Allium genus, Vosa [11] concluded that the position of the C-bands seems to be independent of chiasmata localization. Moreover, COs may be distributed in a more random fashion than chiasmata in the more condensed state of the chromosome at metaphase I. Evidence for this assumption was obtained by analyzing AFLP linkages in *Allium porrum* [71]. It was shown that AFLP markers are not inherited in large linkage blocks, despite the cytogenetic observations of chiasmata occurring mainly in the proximal region. In *A. fistulosum*, the large terminal heterochromatin block consists of a 378 bp satellite repeats interspersed with Ty1-copia retrotransposone sequences and microsatellites [72]. Analysis of the distribution of DNA methylation pattern in *A. fistulosum* using specific antibodies against 5-methylcytosine (anti-5mC) showed highly methylated DNA blocks in the distal regions of all chromosomes [73]. We may suggest that the highly methylated distal heterochromatin blocks move chiasmata away from the telomeric ends in *A. fistulosum*.

Higgins et al. [74], studying the predominantly distal distribution of COs in barley, concluded that the reason for this phenomenon is spatiotemporal asymmetry of the meiotic program. The authors found that the distal euchromatin was the first to replicate, and the formation of DSBs in the distal regions occurred before the elaboration of a continuous chromosome axis in the interstitial/proximal regions. Thus, DSBs in interstitial and proximal regions occur later than distal events, and rarely progresses to form chiasmata. This is in contrast with observations in plants such as Arabidopsis and Brassica, in which the formation of the chromosome axes at late G2/leptotene is very rapid and recombination initiation is synchronous throughout the chromosomes. We may hypothesize that spatiotemporal asymmetry in DNA replication takes place in *A. fistulosum*, then the large terminal blocks of heterochromatin would replicate later than DNA in the interstitial/proximal regions, reducing the ability to form COs in these regions.

It is obvious that epigenetic modification can influence the CO location, as it was demonstrated on *met1* and non-CG/H3K9me2 mutants of Arabidopsis [75,76]. Using whole-genome sequencing of mouse single sperm, providing a genome-wide map of crossovers, genetic and epigenetic factors that affect crossover probability have been identified [77]. An analogue to the single sperm sequencing approach outlined above is an already developed method of the single pollen cell sequencing [78,79], which will allow us to investigate meiotic recombination in plants at the DNA sequence level to directly explore the recombination landscape at the molecular level in an unbiased manner. Further sequencing and telomere-to-telomere assembly of the *A. cepa* and *A. fistulosum* genomes, together with advances in epigenetics, will help resolve this enigma of highly contrasting patterns of chiasmata localization in these closely related Allium species.

### 3.2. The A. cepa Type of Chiasmata Distribution on Homeologous Bivalents in F${}_{1}$ Diploid Hybrid

Despite the difference in the amount of DNA and contrasting chiasmata location between *A. fistulosum* and *A. cepa*, the chromosome conjugation, SC formation and crossing over occurred in the F${}_{1}$ diploid hybrid. Remarkably, all bivalents had predominantly distal and interstitial chiasmata as in *A. cepa*. Our results are in complete agreement with those previously obtained in F${}_{1}$ hybrid between *A. fistulosum* and *A. cepa* [10,46]. It should be noted that we used *A. cepa* as the female parent and *A. fistulosum* as the pollinator, but in previous studies, analysis of bivalents configuration was performed using F${}_{1}$ hybrids, in which the female parent was *A. fistulosum* and the pollinator was *A. cepa*. Thus, the influence of the cytoplasm on the configuration of bivalents can be excluded, at least in the case of these closely related species. Our data confirm the hypothesis of the presence of a dominant gene/genes that provide distal and interstitial COs between the homeologous chromosomes in the F${}_{1}$ diploid hybrid, previously put forward by Emsweller and Jones [10]. However, such an assumption would be very simplistic, given the fact that the hybrid combines two genomes that differ in size [68], as well as intrinsic properties [11,72,73,80].

Using the immunodetection of ASY1 and ZYP1 in prophase I, we revealed a delay in chromosome pairing and the lack of synapsis in several sites between homeologous chromosomes in F${}_{1}$ diploid hybrid. These results agree with data obtained by Albini and Jones [46] on SC spreads using electron microscopy. The authors observed only 80–90% of the synapsed chromosomes at late pachytene in F${}_{1}$ hybrid between *A. fistulosum* and *A. cepa*. It was suggested that the factors involved in promoting the interstitial and distal location of chiasmata might be disturbances of synapsis in the centromeric regions of homeologous bivalents between *A. cepa* and *A. fistulosum* and, probably, this constraint, reduces proximal chiasmata frequency in the F${}_{1}$ diploid hybrid. Using light epifluorescent microscopy, we were not able to determine whether the non-sinapsis sites are located in the pericentromeric regions of the pachytene chromosomes, which look like a tangled ball. Disturbances of synapsis may indicate sites of DNA differences where extra blocks of DNA occur in the *A. cepa* genome. We often observed long ZYP1 tracks that almost coincided with DAPI-stained chromatin. The adjustment of chromosomes axis lengths during synapsis of the parental chromosomes with different lengths in interspecific hybrids could take place [81,82]. However, how the correct homologous sequence is located efficiently among the bulk of chromatin embedded nuclear DNA remains unclear [83].

### 3.3. A Significant Shift in Chiasmata Localization and MLH1/MUS81 Ratio in F${}_{1}$ Triploid Hybrid (2n = 3x = 16F + 8C) Is Evidence of Genetic Control of COs Distribution

Previously, Emsweller and Jones [9] suggested that *A. cepa* possesses a dominant gene, and *A. fistulosum* possesses a recessive one based on an analysis of bivalent configurations in F${}_{1}$ and gene segregation in F${}_{2}$ hybrids between *A. cepa* and *A. fistulosum*. The F${}_{1}$ triploid hybrid, which has a complete diploid set of *A. fistulosum* chromosomes and eight chromosomes of *A. cepa*, is an ideal source for testing the genetic control hypothesis. In the F${}_{1}$ triploid hybrid, bivalents were formed by the *A. fistulosum* homologous chromosomes that have no differences in the length and homology of sequences. Despite this, a significant shift in the localization of chiasmata to the distal and interstitial regions was found (Table 1, Figure 2d). This observation indicates that the genetic control of chiasmata distribution is possible, along with other factors that control crossing over.

Counting chiasmata in bivalents does not allow for determination of the pathway of CO origin. Immunolocalization of MLH1 (class I COs) and MUS81 (class II COs) in *A. cepa*, *A. fistulosum* and their F${}_{1}$ diploid and triploid hybrids showed that the two CO pathways interact. The number of MLH1 foci per cell is not independent of the number of MUS81 foci (Table 3). Moreover, the MLH1/MUS81 ratio correlates with the distribution of chiasmata. We observed that a higher MLH1/MUS81 ratio was linked with an increase in the number of distal and interstitial chiasmata. In the F${}_{1}$ triploid hybrid, a significant increase in MLH1 foci per cell compared to *A. fistulosum* was observed. A more pronounced difference in the MLH1/MUS81 ratio was found in the parental species. With this in mind, it was of interest to evaluate whether the protein sequences of MLH1 and MUS81 differ in *A. cepa* and *A. fistulosum*. We found that MLH1 is highly conserved between *A. cepa* and *A. fistulosum*. Transcripts of MLH1 (*A. cepa*: GBRO01051308.1, *A. fistulosum*: GFAM01047024.1) contains ORF (Open Reading Frame) encoding proteins of equal length (725 amino acids) and share 99.9% (724/725) amino acids identity with polymorphism in one residual. Thus, the *mlh1* genes of *A. cepa* and *A. fistulosum* should produce almost identical proteins. In the F${}_{1}$ triploid hybrid, MLH1 could be involved in recombination at an increased dosage if all three loci (two *A. fistulosum* and one *A. cepa*) express in equal proportions. However, in newly synthesized interspecific hybrids, a genome-wide expression level dominance and/or homeolog expression bias depending on species and/or tissue was shown [84].

Increasing the number of MLH1 foci per cell, the number of MUS81 foci per cell was significantly decreased in the F${}_{1}$ triploid hybrid compared to *A. fistulosum*, that is consistent with the statement of COs homeostasis to maintain a fixed amount of COs per cell [18,85]. Interestingly, MUS81 is more polymorphic between *A. cepa* and *A. fistulosum* compared to MLH1. Transcripts of MUS81 (*A. cepa*: GBRQ01006735.1, *A. fistulosum*: GHMM01233510.1) contains ORF encoding proteins of equal length (604 amino acids) and share 93.5% (565/604) amino acids identity with polymorphism in 39 residuals with increasing identity in winged helix domain (96.2%; 102/106) and XPF-nuclease domain (99.3%; 145/146). MUS81 protein sequences showed considerable interspecific polymorphism between *A. cepa* and *A. fistulosum* even in functional domains. Whether this structural polymorphism of the MUS81 is associated with possible functional features and the dominant/recessive allelic stage is unknown.

In summary, we have shown, for the first time, (1) a significant shift in chiasmata localization on the *A. fistulosum* homologous chromosome pairs in the presence of haploid set of the *A. cepa* chromosomes in F${}_{1}$ triploid hybrid, and (2) an increase in the number of MLH1 foci per cell was balanced by a decrease in the number of MUS81 foci in the F${}_{1}$ triploid hybrid, which is consistent with CO homeostasis, (3) a significant difference in the MLH1/MUS81 ratio between *A. cepa* and *A. fistulosum*, and (4) in the F${}_{1}$ diploid hybrid, the MLH1/MUS81 ratio was similar to that of *A. cepa*, and (5) there is no difference in assembly and disassembly of ASY1 and ZYP1 between *A. cepa* and *A. fistulosum*, which was similar to that of maize, wheat and rice, and (6) the ZYP1 assembly delay was found in the F${}_{1}$ diploid hybrid. Finally, our results support the hypothesis of genetic control of CO localization. Given the biological importance and complexity of crossing over, the control of CO localization must be considered on many levels beyond genetics, including epigenetics, as the template’s chromatin environment, spatiotemporal DNA replication and the distribution of DSBs, impact of the protein modification and interaction, and others.

## 4. Materials and Methods

### 4.1. Plant Materials

Perennial plants of *A. fistulosum* (2*n* = 2*x* = 16), of the Ruskiy Zimniy variety grow on the experimental field of the Center for Molecular Biotechnology RSAU-MTAA. Bulbs of *A. cepa* (2*n* = 2*x* = 16), of the Chalcedon variety were planted in the field. F${}_{1}$ hybrids between *A. cepa* and *A. fistulosum* were obtained by manual emasculation of anthers in unopened buds. Pollen from *A. fistulosum* plants were collected in Petri dishes and stored at +4 ${}^{\circ }$C until the pistil stigma ripened for pollination in emasculated buds of *A. cepa*. Pollination of each flower was carried out for 2 days by reapplying pollen on the stigma of the pistil with a brush. F${}_{1}$ diploid hybrid (*A. cepa*×*A. fistulosum*) (2*n* = 2*x* = 8F + 8C) acc. 1–20 and F${}_{1}$ triploid hybrid (*A. cepa*× *A. fistulosum*) (2*n* = 3*x* = 16F + 8C) acc. 7–20 were used in this study.

### 4.2. Genomic In Situ Hybridization (GISH)

Mitotic chromosomes of F${}_{1}$ hybrids were prepared from young root meristems using the squash method according to Kudryavtseva et al. [86]. *In situ* hybridization, immunological detection, and counterstaining procedures were the same as previously described by Khrustaleva and Kik [87]. The hybridization mixture contained: 50% (*v*/*v*) formamide, 10% (*w*/*v*) dextran sulfate, 2 × SSC, 0.25% (*w*/*v*) sodium dodecyl sulfate (SDS), 50 ng/μL of labeled probe (DIG)-11-dUTP and 1500 ng/μL of blocking DNA. In the hybridization mixture, we used a ratio of 1:30 of probe and block DNA, and washes at 78% stringency were applied. In F${}_{1}$ (*A. cepa*×*A. fistulosum*) (2*n* = 2*x* = 8F + 8C) the *A. fistulosum* genomic DNA was used as a probe and the *A. cepa* genomic DNA was used as a block. In F${}_{1}$ (*A. cepa*× *A. fistulosum*) (2*n* = 3*x* = 16F + 8C) the *A. cepa* genomic DNA was used as a probe and the *A. fistulosum* genomic DNA was used as a block. In both cases, the labeled (DIG)-11-dUTP probe DNA was detected with anti-Dig-FITC raised in sheep (Roche diagnostics GmbH, Mannheim, Germany) and amplified with anti-sheep-FITC raised in rabbit (Vector Laboratories, CA, USA).

GISH preparations were visualized using a Zeiss AxioImager M2 microscope (www.zeiss.com, accessed on 15 November 2021) and a black–white sensitive digital Hamamatsu camera C13440-20CU (www.hamamatsu.com, accessed on 15 November 2021). The final optimization of images was performed using Adobe Photoshop (Adobe Inc., San Jose, CA, USA). Karyotype analysis and identification of individual chromosomes with fluorescent signals by the DRAWID program [88] were performed according to bulb onion nomenclature [89] and previously published karyotypes of closely related Allium species [90].

### 4.3. Analysis of the Chiasmata Distribution in PMCs

Fresh unfixed anthers containing pollen mother cells (PMCs) at metaphase I were tapped out in a drop of 1% acetocarmine on a glass slide, gently mixed, and then heated for 1 min at 60 ${}^{\circ }$C on a heating table. The cells were spread by tapping on the coverslip and gently squashed. Slides were examined under a Zeiss Axiolab 5 microscope (www.zeiss.com, accessed on 28 August 2021) using phase-contrast microscopy. The selected images were captured using a digital Axiocam 208 color camera (www.zeiss.com, accessed on 28 August 2021). Image processing was performed by using Zen 2.6 lite (blue edition), an image analysis software. Bivalent arms were arbitrarily divided into three regions of equal length (proximal, interstitial, distal). The position of chiasmata on chromosome arm was measured by a DRAWID program [88]. Only non-overlapping bivalents were used for measurements of positions of chiasmata.

### 4.4. Immunochemical Analysis

#### 4.4.1. Antibody Production

Reference transcriptomes of *A. cepa* and *A. fistulosum* from GenBank (Transcriptome Shotgun Assembly) were used for the acquisition of ZYP1, MLH1 and MUS81 protein sequences. Sequences of these proteins from other species in order to use as a reference were identified based on literature data: ZYP1 (*Zea mays*, GenBank: ADM47598.1, [54]), MLH1 (*Solanum lycopersicum*, GenBank: EF071927.1, [41]), MUS81 (*Arabidopsis thaliana*, GenBank: AB177892.1, [91]). tBLASTn was used for identification transcript sequences of the corresponding proteins in Transcriptome Shotgun Assembly database (identity >50%, query cover > 70%). The longest Open Reading Frame (ORF) was translated into an amino acid sequence. The following steps, including identification of a peptide sequence for antibody production and antibody synthesis, were performed by the outsource company PrimeBioMed (www.primebiomed.ru, accessed on 9 February 2020). The anti-ASY1 polyclonal antibody (raised in rabbit against ASY1 of *Arabidopsis thaliana*) was described by Armstrong et al. [23].

#### 4.4.2. Preparation of Meiotic Chromosome for Protein Immunolocalization

We developed an original method of meiotic chromosome preparation. Anthers at the desired stage of meiosis were fixed by Clark’s fixative (ethanol:acetic acid, 3:1, *v*/*v*) for the 1 h. The fixed anthers were washed in tap water for 30 min, and then in citrate buffer (10 mM sodium citrate, 10 mM citric acid, pH 4.8) for 10 min. A total of 5–8 anthers were transferred into 1.5 mL tubes with 50 μL of 0.6% enzyme mixture (1:1:1) pectolyase Y-23 (Kikkoman, Tokyo, Japan), Cellulase Onozuka R-10 (Yakult Co., Ltd., Tokyo, Japan) and Cytohelicase (Sigma-Aldrich Co., LLC, St. Louis, MO 63103 USA) for 120 min at 37 ${}^{\circ }$C. After 120 min, the enzyme mixture was deleted by Pasteur Pipettes from the tubes. Anthers were gently transformed into a fine cell suspension in the tube using a dissecting needle. A total of 100 μL of 60% acetic acid was added to the cell suspension. The tube with cell suspension was then heated for 5 min at 50 ${}^{\circ }$C. A total of 60 μL of ethanol/acetic acid fixative at a ratio of 3:1 was added. A total of 10 μL of cell suspension was dropped onto a slide, and by the time the surface became granule-like (10–15 s), 30 μL of ethanol/acetic acid fixative at a ratio of 3:1 was added. Then the slides with cells were immediately dried with an airflow. The same day, the slides were used for immunochemical analysis. Some slides were kept in the fridge—70 ${}^{\circ }$C.

#### 4.4.3. Immunochemical Detection of ASY1 and ZYP1

Slides were incubated with blocking buffer (5% BSA in 1 × PBS, 0.1% Tween-20, 1 mM EDTA, pH = 8.0) for 2 h at room temperature. Then, slides were incubated with rabbit anti-ASY1 (1:20) and rat anti-ZYP1 (1:20) in the blocking buffer for 24 h at 4 ${}^{\circ }$C. Slides were washed 4 × 30 min in 1 × PBS, 0.1% Tween-20, 1 mM EDTA (pH = 8) at room temperature. Slides were incubated with blocking buffer for 2 h at room temperature. The following secondary antibodies diluted in block buffer (1:100) were applied at 37 ${}^{\circ }$C for 1 h: for ASY1, the visualization was goat anti-rabbit Alexa Fluor 555, (red fluorescence) (Abcam plc, Cambridge, CB2 0AX, UK), and for ZYP1 it was goat anti-rat Alexa Fluor 488, (green fluorescence) (Abcam plc, Cambridge, CB2 0AX, UK). The slides were washed 4 × 30 min in 1 × PBS, 0.1% Tween-20, 1 mM EDTA (pH = 8) at room temperature and mounted in Vectashield antifade medium (Vector Laboratories) with 2 μg/mL DAPI.

#### 4.4.4. Immunochemical Detection of MLH1/ZYP1 and MUS81/ZYP1

The incubation and washing steps were the same as described above. The MLH1 immunodetection was with mouse anti-MLH1 (1:100), followed by the secondary goat anti-mouse antibody conjugated with Alexa Fluor 555, (red fluorescence) (Abcam plc, Cambridge, CB2 0AX, UK). The MUS81 immunodetection was with guinea pig anti-MUS81 (1:100), followed by the secondary goat anti-guinea pig Alexa Fluor 555 (red fluorescence) (Abcam plc, Cambridge, CB2 0AX, UK). Simultaneous ZYP1 immunodetection was with rat anti-ZYP1 (1:20), followed by goat anti-rat Alexa Fluor 488 (green fluorescence) (Abcam plc, Cambridge, CB2 0AX, UK).

#### 4.4.5. Statistical Analysis

Statistical analysis of chiasmata "location types" distribution was performed applying $\chi^{2}$-statistics from a stats package using R programming language v4.2.1 [92]. Pairwise comparisons between *A. cepa*, *A. fistulosum* and F${}_{1}$ diploid and triploid (*A. cepa*×*A. fistulosum*) hybrids chiasmata “location types” was performed using $\chi^{2}$-statistics. *p* values were adjusted using Bonferroni correction. Effect size (Cramer’s V) was calculated using a cramerV function from the rcompanion v2.4.21 package [93].

Identification of statistically significant differences in MLH1 and MUS81 foci number per cell among *A. cepa*, *A. fistulosum*, and F${}_{1}$ diploid and triploid hybrids was performed using the Wilcoxon test in R programming language v4.2.1 [92]. Calculated *p* values were adjusted using the False Discovery Rate (FDR). A *p* value threshold < 0.05 was considered as statistically significant.

An estimation of the mean MLH1/MUS81 ratio per cell was performed via generation of all possible pairwise combinations of MUS81 and MLH1 foci (artificial population) in analyzed cells for each analyzed species (Appendix A). For each pair, the MLH1/MUS81 ratio was calculated. The weighted mean with inverse weights was calculated for each analyzed species in order to decrease the magnitude of influence of combinations with an extreme MLH1/MUS81 ratio.

#### 4.4.6. Microscopy and Imaging

The slides were examined under a Zeiss AxioImager M2 microscope (www.zeiss.com, accessed on 15 November 2021). The selected images were captured using a digital Hamamatsu camera C13440-20CU (www.hamamatsu.com, accessed on 15 November 2021). Image processing was performed by Zen 2.6 (blue edition), an image analysis software.

## Figures and Tables

**Figure 1 ijms-24-07066-f001:**
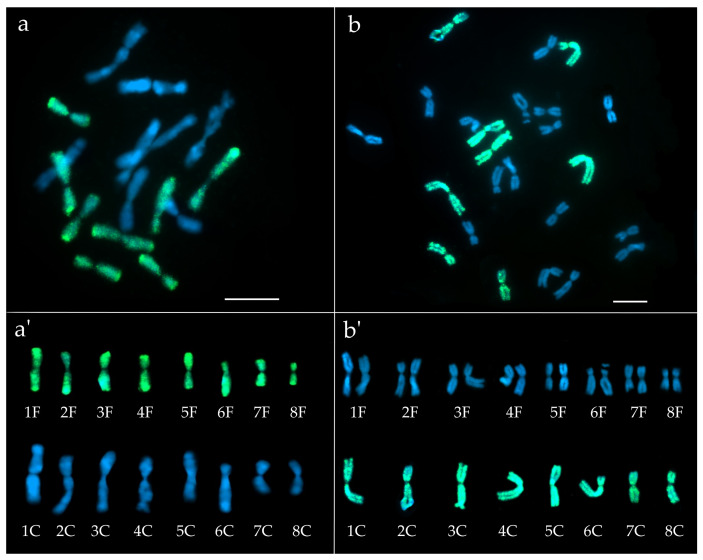
GISH on mitotic metaphase chromosomes of F${}_{1}$ interspecific hybrids between *A. cepa* and *A. fistulosum*: (**a**) F${}_{1}$ diploid hybrid contains 8 chromosomes of *A. cepa* and 8 chromosomes of *A. fistulosum*; (**b**) F${}_{1}$ triploid hybrid contains 8 chromosomes of *A. cepa* and 16 chromosomes of *A. fistulosum*; (**a**${}^{\prime }$,**b**${}^{\prime }$) karyotypes of the F${}_{1}$ diploid and F${}_{1}$ triploid hybrids, respectively. Bars represent 10 μm.

**Figure 2 ijms-24-07066-f002:**
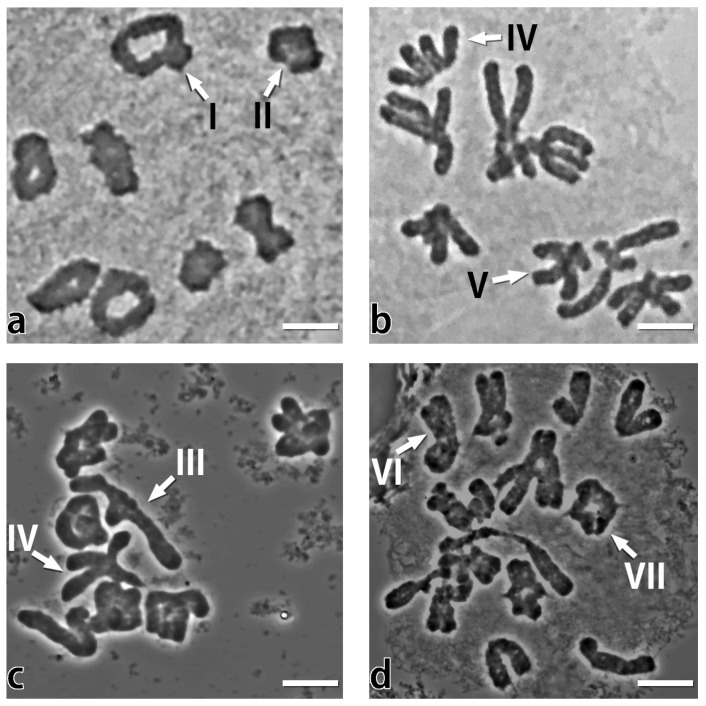
Acetocarmine-stained squash preparations of PMCs at metaphase I: (**a**)—*A. cepa*, (**b**)—*A. fistulosum*, (**c**)—F${}_{1}$ (*A. cepa*×*A. fistulosum*) diploid hybrid (2*n* = 2*x* = 8F + 8C), (**d**)—F${}_{1}$ (*A. cepa*×*A. fistulosum*) triploid hybrid (2*n* = 2*x* = 16F + 8C). Roman numbers mark different types of bivalents: I—ring bivalent with two distal and single interstitial chiasmata, II—ring bivalent with two distal chiasmata, III—heteromorphic open bivalent with interstitial chiasma, IV—univalent, V—cross-bivalent, VI—H-form of cross-bivalent, VII—ring bivalent with two interstitial chiasmata. Bars represent 10 μm.

**Figure 3 ijms-24-07066-f003:**
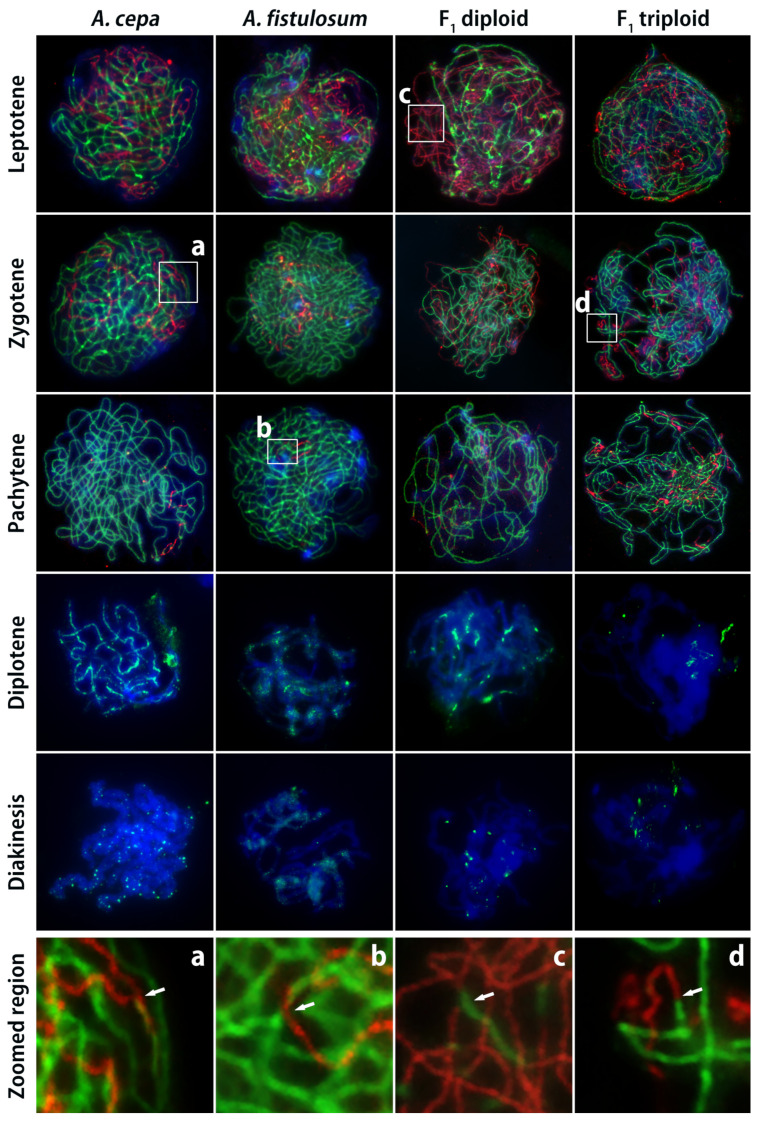
The behavior of ASY1 (red) and ZYP1 (green) during prophase I in *A. cepa*, *A. fistulosum* and their F${}_{1}$ diploid and triploid hybrids. Chromatin was stained with DAPI. (**a**–**d**) Arrows in zoomed regions are pointing at regions with absence of the ASY1 signal after loading ZYP1.

**Figure 4 ijms-24-07066-f004:**
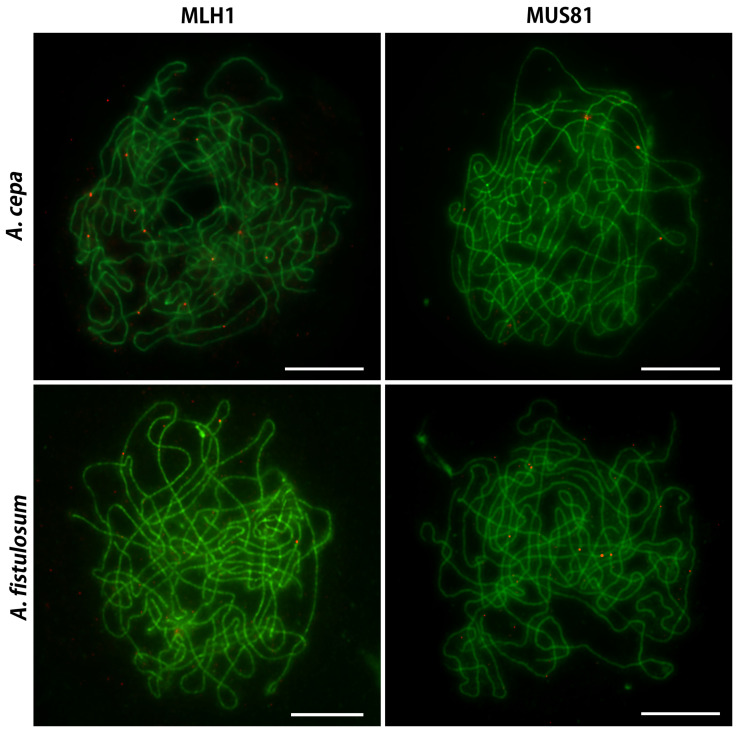
Immunolocalization of the MLH1/ZYP1 or MUS81/ZYP1 at pachytenes of *A. cepa* and *A. fistulosum*. MLH1 (on the **left**) marks class I COs, and MUS81 (on the **right**) marks class II COs. Bar = 10 μm.

**Figure 5 ijms-24-07066-f005:**
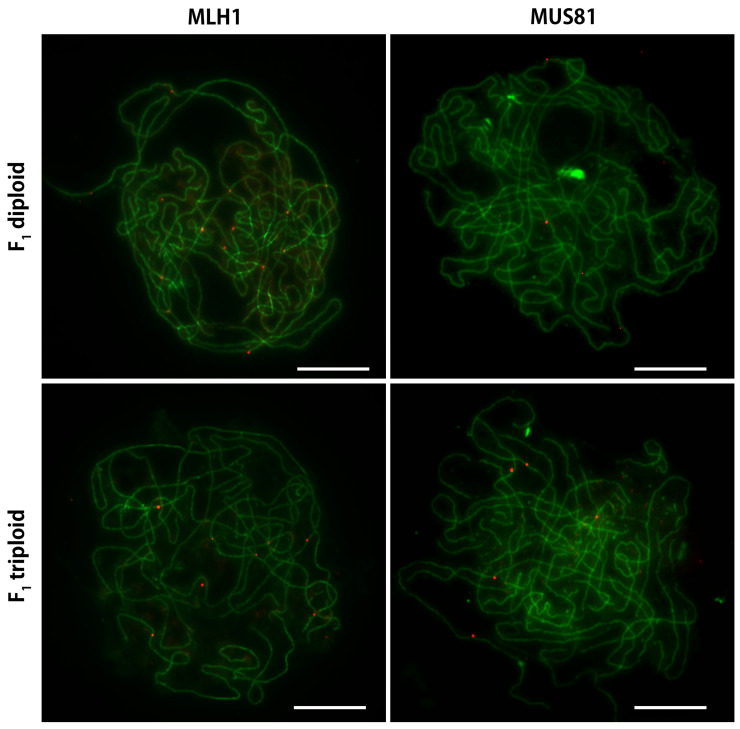
Immunolocalization of the MLH1/ZYP1 and MUS81/ZYP1 at pachytenes of F${}_{1}$ diploid hybrid (2*n* = 2*x* = 8F + 8C) and F${}_{1}$ triploid hybrid (2*n* = 3*x* = 16F + 8C). Bar = 10 μm.

**Table 1 ijms-24-07066-t001:** Chiasmata frequency and distribution in *A. cepa*, *A. fistulosum*, and the F${}_{1}$ diploid and triploid hybrids (*A. cepa*× *A. fistulosum*).

	Number of Cells Scored	Mean PMC Chiasma Frequency	Number of Bivalents Analyzed	Chiasmata Location		
				Proximal % (Total)	Interstitial % (Total)	Distal % (Total)
*A. cepa*	54	19.1	435	1.8 (18)	20.0 (207)	78.2 (808)
*A. fistulosum*	43	15.4	340	97.1 (643)	1.2 (8)	1.7 (11)
F${}_{1}$ (*A. cepa* × *A. fistulosum*)						
(2*n* = 2*x* = 8F + 8C)	35	12.0	248	11.4 (48)	43.0 (180)	45.6 (191)
F${}_{1}$ (*A. cepa* × *A. fistulosum*)						
(2*n* = 3*x* = 16F + 8C)	29	14.6	231	39.5 (167)	26.0 (110)	34.5 (146)

**Table 2 ijms-24-07066-t002:** Pairwise comparisons of chiasmata type distribution in *A. cepa*, *A. fistulosum*, and the F${}_{1}$ diploid and triploid hybrids (*A. cepa*× *A. fistulosum*).

Pairwise Comparison	Cramer’s V	$\chi^{2}$*p* Value *
*A. cepa* vs. *A. fistulosum*	0.95	<0.001
*A. fistulosum* vs. F${}_{1}$ diploid hybrid	0.87	<0.001
*A. fistulosum* vs. F${}_{1}$ triploid hybrid	0.72	<0.001
*A. cepa* vs. F${}_{1}$ triploid hybrid	0.49	<0.001
*A. cepa* vs. F${}_{1}$ diploid hybrid	0.34	<0.001
F${}_{1}$ diploid hybrid vs. F${}_{1}$ triploid hybrid	0.24	<0.001

* Bonferroni corrected.

**Table 3 ijms-24-07066-t003:** The frequency of MLH1 and MUS81 signals separating on ZYP1 tracks at the pachytene of *A. cepa*, *A. fistulosum*, and the F${}_{1}$ diploid and triploid hybrids (*A. cepa*× *A. fistulosum*).

	MLH1				MUS81			∑ Foci per Cell *
	Total Cells	Total Signals	Mean Signals per Cell		Total Cells	Total Signals	Mean Signals per Cell	
*Allium cepa*	37	409	11.1 (73%)		39	158	4.1 (27%)	15.2 (100%)
*Allium fistulosum*	40	257	6.5 (50%)		41	272	6.5 (50%)	13.0 (100%)
F${}_{1}$ (*A. cepa* × *A. fistulosum*)								
(2*n* = 2*x* = 8F + 8C)	38	320	8.5 (70%)		37	136	3.6 (30%)	12.1 (100%)
F${}_{1}$ (*A. cepa* × *A. fistulosum*)								
(2*n* = 3*x* = 16F + 8C)	36	311	8.5 (60%)		38	214	5.7 (40%)	14.2 (100%)

* ∑—The sum of the average values per cell

## Data Availability

Data available in a publicly accessible repository.

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
