# Peer review of "The Control of the Crossover Localization in Allium"

_ijms, 2023, doi:10.3390/ijms24087066_

Round 1

Reviewer 1 Report

The manuscript is aimed at a comparative analysis of chiasma distribution in two closely related onion species with contrast recombination pattern (Allium cepa and A. fistulosum) and in their hybrids. The authors use the method of chiasma count, immunolocalization of meiotic proteins, and transcription analysis of the MLH1 and MUS81 which mark two classes of crossover events.

The main benefit of this article is the use of an unique model - the triploid hybrids with diploid chromosome set of A. fistulosum (known to have proximal chiasma localization) and a haploid set of A. cepa (characterized by more even distribution). The authors have convincingly shown the distalization of the recombination on bivalents derived from A. fistulosum in the presence of the haploid chromosome set of A. cepa. This finding strongly supports the hypothesis of the genetic control of the CO distribution and serves as an important step towards the understanding how crossover positioning is regulated.

Unfortunately, this is the only advantage of the article. The elegant idea of comparing the number and distribution of MLH1 and MUS81 foci to check if the contrasting CO pattern is related to the different CO repair pathways (this goal, by the way, is not formulated anywhere in the article) has not been properly implemented. The total number of MLH1 and Mus81 sites was about 2 times less than expected based on the chiasma analysis. Authors explain this by the very compacted chromatin. However, how could one make the statistically sound conclusions based on the incomplete set of the data (which is obviously not representative since the chromatin compaction and hence signal detection is not uniform)? Moreover, the number of cells studied (8) is hardly enough to draw the conclusions, especially since there is a pronounced background for MLH1 staining seen at Figure 8 (especially for A. cepa). It is also not possible to determine the beginnings and ends of the chromosomes to define the CO position from the presented pictures.

The authors also do not suggest any motivation for the experiments on mlh1 and mus81 transcription measuring. Probably the authors consider them as candidate genes for the genetic control of CO distribution. At least, this can be judged from the phrase in the Discussion: "Based on our result, we expect that in alien monosomic addition line with chromosome 2 [where mlh1 gene is located] of A. cepa the number of proximal chiasmata will be significantly decreased" (Lines 354-355). This assumption is more than premature, not only because no association analysis was made, but also because MLH1 is involved in the late stage of DSB processing and is unlikely to be related to its positioning. For that reason author's assumption that the maximum expression of mlh1 in leptotene in hybrids may be related to "its involvement in DNA repair" (Lines 437-438) looks strange.

The text is replete with grammatical, stylistic errors, inconsistencies and is difficult to understand. Not only the language required revising but also the presentation of the material.

Here are some of the points, which require improvement:

-it is not clear what the Figure 3 with the correspondence analysis serves for;

-showing four very similar figure panels (Figures 4-7) is redundant to illustrate a minor difference, some of them might be placed in the Supplementary; 

-the part of the paragraph 2.4 at the p.8 is devoted to the pairwise comparison of 8 numbers in a style that is unacceptable for the scientific papers (even for the bachelor thesis);

-the same is true for the description of transcription results;

-showing two types of graphical representation using the same data (Table 1 and Figure 3; Table 2 and Figure 9) must be avoided; 

-it is not clear what authors mean by "spatiotemporal asymmetry among A. cepa, A. fistulosum" (abstract) showed by immunolocalization and gene expression profiling of MLH1 and MUS81.

-not all features in the Figure 2 are reflected in the caption.

In summary, despite the scientific value of some findings, not all results are reliable and properly interpreted. The manuscript is littered with unnecessary repetitions, the language requires a complete revision. The presentation of the results is not acceptable and the manuscript cannot be published in its current form.

Reviewer 2 Report

Dear Authors

kindly find the attached and correct the minor mistakes highlights

Reviewer 3 Report

The manuscript submitted to International Journal of Molecular Science developed by Kudryavtseva et al. They described meiotic crossovers/chiasmata in species of the genus Allium. After reviewing it, I would consider the following points:

i. They use individual plants with 16 and 24 chromosomes obtained by crossing of A. cepa (as female) and A. fistulosum (as male). However, it is not clear why two types of F1 hybrids are developed (one with 16 chromosomes and other with 24 chromosomes) as well as the triploid hybrid. If these lines were developed in this work, author should include the development with the different lines obtained. The effect of the A. cepa haploid genoma on the configuration of the A. fistulosum. Why is this relevant?

ii. All the A. cepa chromosomes are observed as bivalent in metaphase I (Figure 2a). The A. fistulosum chromosomes are not observed in the same chromosomal rearragenment. These chromosomes showed chiasmata to the proximal regions (mainly). Is this observed in other plant species? From my experience, ring bivalents are commonly observed in meiotic metaphase I.

iii. Lines 142-146. Apart form the different genoma size, one way to know exactly the chiasma formation between both Allium species is to carry out GISH in the meiotic metaphase I. I recommend showing them in the manuscript. In the figure 2, the green arrow is not described. The Pink arrows of figure 2a are not correctly described in the legend. I would say that the chromosomal rearrangement showed by the blue arrow in the figure 2c is a distal chiasma, this is a rod bivalent. More cells about figure 2b should be include to confirm the chromosomal rearrangements. Legend of figure 2 should be clarified.

iv. If the figures 4-7 are joined by meiotic stage rather than by species, maybe, they help to compare them easily.

v. Lines 200-201 are not relevant here.

vi. The signals of MLH1 and MUS81 are not clear in the figure 8. They are not appreciable.

vii. Lines 364-365 include information that are expectations that are not related to the contento of the manuscript.

viii. Lines 381-382.  Is it only observed in these species? Discuss other species.

ix. Lines 383-384. Why the high degree of condensation of Allium chromosomes affect to the results?

x. Line 401. Why a limited number of cells available for analysis was used?

Round 2

Reviewer 1 Report

In the revised form, the manuscript "The control of the crossover localization in Allium" was noticeably improved. However, it cannot be published in its current form. Despite the noticeable reworking, the manuscript is still rather negligent. The English is still in need of significant revision. In some cases, the terms are used improperly. The text contains a lot of redundancy and overspeculation. The data from tables are verbosely duplicated in the text. The results section contains a lot of speculation that is duplicated in the discussion section.

There are some points to revise:

1) How does the epigenetic control of CO distribution follow from the results of the work? Moreover, the authors note that there were no significant differences in chromatin organization between the species.

2) Line 32, 303. Obligate crossing over is by definition one per bivalent, not 1-2;

3) There are contradicting sentences in lines 66-69;

4) Lines 125-128. the sentence is out of place. It has nothing to do with the following text;

5) Line 137. It is not clear why authors claim that the expected frequency of chiasmata per cell is 16;

6) It is not clear what authors mean in the sentence in lines 215-216;

7) Figure 5 caption is mixed up;

8) Line 351. I suspect that not all of the 369,054 vascular plant species have been examined for recombination distribution. It is more correct and informative to discuss to discuss the occurrence of the trait among the studied species;

9) The text contains sentences with poor grammar, misspelling and inconsistencies. There are some examples to revise: lines 35-36, 41-42, 46-48, 66-68, 103, 179-182, 241-242, 392-394, 401, 406, 424, 452-455, 480, 482.

Reviewer 3 Report

After considering the revised version of the manuscript, it has been improved but it is not enough. The following comment was not replied:

iii. Lines 142-146. Apart form the different genome size, one way to know exactly the chiasma formation between both Allium species is to carry out GISH in the meiotic metaphase I. I recommend showing them in the manuscript.  

The number of vascular plant species included in the following sentence:

“To our knowledge, only four plant species among 369,054 vascular plant species are known with such extreme localization of chiasmata, namely, Allium fistulosum [10], Allium porrum [69,70], Allium altaicum [11,71], which are very close phylogenetically, and Fritillaria meleagris [72] belonging to the Liliaceae family. “ It is quite ambitious.

The figure 8 with signals of MLH1 and MUS81 has been removed because it was not clear. However, in the current version, the number of cells is much higher and the quality of pics is better. How has it been improved?

Moreover, significant revisions still need to be made to English. The results and discussion are quite similar and the degree of expeculation is too high.

Round 3

Reviewer 3 Report

N/A